# Identification and Expression Patterns of *WOX* Transcription Factors under Abiotic Stresses in *Pinus massoniana*

**DOI:** 10.3390/ijms25031627

**Published:** 2024-01-28

**Authors:** Dengbao Wang, Zimo Qiu, Tao Xu, Sheng Yao, Mengyang Zhang, Xiang Cheng, Yulu Zhao, Kongshu Ji

**Affiliations:** State Key Laboratory of Tree Genetics and Breeding, Key Open Laboratory of Forest Genetics and Gene Engineering of National Forestry and Grassland Administration, Co-Innovation Center for Sustainable Forestry in Southern China, Nanjing Forestry University, Nanjing 210037, China; dbw@njfu.edu.cn (D.W.); zmqiu@njfu.edu.cn (Z.Q.); w2110103115@njfu.edu.cn (T.X.); smilence@njfu.edu.cn (S.Y.); younggang@njfu.edu.cn (M.Z.); chengxiang@njfu.edu.cn (X.C.); ylzhao@njfu.edu.cn (Y.Z.)

**Keywords:** WOX, *Pinus massoniana*, abiotic stress, expression patterns

## Abstract

WUSCHEL-related homeobox (WOX) transcription factors (TFs) play a crucial role in regulating plant development and responding to various abiotic stresses. However, the members and functions of WOX proteins in *Pinus massoniana* remain unclear. In this study, a total of 11 WOX genes were identified, and bioinformatics methods were used for preliminary identification and analysis. The phylogenetic tree revealed that most PmWOXs were distributed in ancient and WUS clades, with only one member found in the intermediate clade. We selected four highly conserved *WOX* genes within plants for further expression analysis. These genes exhibited expressions across almost all tissues, while *PmWOX2*, *PmWOX3*, and *PmWOX4* showed high expression levels in the callus, suggesting their potential involvement in specific functions during callus development. Expression patterns under different abiotic stresses indicated that *PmWOXs* could participate in resisting multiple stresses in *P. massoniana*. The identification and preliminary analysis of *PmWOXs* lay the foundation for further research on analyzing the resistance molecular mechanism of *P. massoniana* to abiotic stresses.

## 1. Introduction

Pinaceae plants represent the largest group of tree species among extant gymnosperms, demonstrating strong adaptability in global ecosystems. They play a significant role in carbon-oxygen balance and possess immense economic value [1]. *Pinus massoniana* Lamb. stands out for its remarkable adaptability, and is commonly known as a widespread cultivation tree in southern provinces of China. It serves as a major source of solid wood and pulp production in China. Moreover, the majority of resin and turpentine oil is also produced by *P. massoniana* [2]. However, its growth and resin yield are limited by deteriorating climate, poor growing conditions, and forest pests and diseases. It is very important to analyze the resistance mechanism to various adversities in *P. massoniana*.

The WOX (WUSCHEL-related homeobox) transcription factor family exists exclusively in plants, and constitutes a subfamily of the homeobox (HB) protein superfamily [3]. The WOX family is characterized by a highly conserved domain ‘helix-loop-helix-turn-helix’ structure, which is composed of 60–66 amino acids called the homeodomain (HD). The HD is encoded by specific DNA sequences that regulate target gene expression at specific times during plant development [4]. In addition to the HD, all WOX members contain a WUS-box motif that regulates stem cell properties and floral meristem size. WOX proteins containing this motif may act as transcriptional repressors in plants [5]. The plant WOX transcription factor family can be categorized into three groups: the WUS clade, the intermediate clade, and the ancient clade [3]. The ancient clade is composed of WOX members from some green algae and land plants, while the WUS clade is made up of seed plants [6]. Researchers have identified WOX families in a variety of plants in recent years. The *WOX* gene family in *Arabidopsis* consists of 15 members. *AtWOX1-7* and *AtWUS* were clustered in the WUS clade, *AtWOX8*, *9*, *11*, and *12* were in the intermediate clade, and *AtWOX10*, *13*, and *14* in the ancient clade [7]. The genome of *P. pinaster* contains 14 *WOX* genes, while *P. taeda* has 11 [8]; *Oryza sativa* contains 13 *WOX* genes [9]; *Zea mays* contains 21 *WOX* genes [10]; *Populus trichocarpa* contains 12 real *WOX* genes, and *Sorghum bicolor* contains 11 *WOX* genes [11]; *P. tabulaeformis* contains 12 *WOX* [12], *Picea abies* contains 12 *WOX* [13], *Ginkgo biloba* contains seven *WOX* genes and *P. sylvestris* contains four *WOX* genes [14], and *Triticum aestivum* L. contains 14 *WOX* genes [15]. In upland cotton, such as *Gossypium arboreum*, *G. raimondii*, and *G. hirsutum*, there are, respectively, reported counts of 21, 20, and 38 *WOX* genes [16].

Previous studies have reported that *WOX* genes are widely involved in plant responses to environmental stresses like drought, salt, cold, etc. *AtHOS9* (*AtWOX6*) can positively regulated freezing tolerance in *Arabidopsis* [17]. Overexpression of *OsWOX11* can control root development and improve the tolerance of rice to potassium deficiency [18]. The overexpression of *OsWOX13,* driven by the rab21 promoter, enhances drought tolerance and accelerates flowering by 7–10 days in rice. *OsWOX13* directly binds to the ATTGATG motif in the promoters of drought-responsive genes *OsDREB1A* and *OsDREB1F*, thus mediating rice’s response to drought stress [19]. *PagWOX11/12a* can regulate the salt tolerance of poplar by activating the expression of *PagCYP736A12* gene [20]. Moreover, *PagWOX11/12a* can directly bind to *SAUR36* and significantly promote adventitious root development and increased salt tolerance in poplar [21]. The expression of *PagWOX11/12a* was predominantly observed in the roots of the ‘84K’ poplar (*Populus alba* × *P. glandulosa*) and exhibited significant upregulation under drought stress conditions. Compared to the wild type, transgenic poplar that overexpressed *PagWOX11/12a* exhibited increased root biomass and improved drought tolerance. Conversely, poplar with inhibited *PagWOX11/12a* exhibited a contrasting phenotype [22]. Additionally, *PagWOX11/12a* can regulate the level of reactive oxygen species (ROS) through modulation of ROS-related gene expression to enhance plant drought resistance [23]. Overexpression of *JrWOX11* in poplar enhances tolerance of NaCl and osmotic stress in transgenic poplar [24]. Overexpression of *JcWOX5* (*Jatropha curcas*) in rice has been shown to enhance plant susceptibility to drought stress [25]. *GhWOX4* can enhance the tolerance of drought stress by regulating the growth of the vascular system in cotton (*Gossypium Hirsutum*) [26].

Although the identification and function of WOX genes have been extensively studied in various species, no reports on *P. massoniana* are available. Therefore, we identified *PmWOX* genes and conducted preliminary analysis using bioinformatics methods. Subsequently, we analyzed the expression patterns of four selected WOX genes across various tissues and treatments. Moreover, we detected their transcriptional activation. This study provides insights into the role of WOX genes under abiotic stresses.

## 2. Results

### 2.1. Identification and Cloning of WOX Genes in P. massoniana

We identified eight putative *WOX* genes from four *P. massoniana* transcriptomes in the absence of a reference genome and designated them in the order of identification as WOX followed by numbers in ascending order: *PmWOX1*-*8*. Moreover, we observed a high degree of conservation in the *WOXX*/*WOX8*, *WUS*, and *WOX3*/*PRS1* sequences within the genomes of *P. tabulaeformis*, *P. pinaster*, and *P. taeda.* However, despite our efforts to blast these three genes in various transcriptomes, no significant matches were found. To investigate further, we extracted RNA from different tissues and reverse transcribed it into cDNA for use as PCR templates. Surprisingly, our PCR experiments consistently failed to clone these three genes using cDNA derived from needle, stem, or root tissues. Ultimately, we were able to clone these three genes using templates obtained from 10-day-old seedlings (Figure 1). We designated these three genes as *PmWOX9*, *PmWUS*, and *PmWOXX*, with lengths of 567 bp, 648 bp, and 506 bp, respectively. Through multiple sequences alignment, we found that these three genes are highly conserved within the Pinaceae family. The homology between *PtPRS1* and *PmWOX9*, *PtWOX7* and *PmWOXX*, and *PtWUS* and *PmWUS* is 98.26%, 98.63%, and 98.3%, respectively. We have submitted *PmWOX9*, *PmWUS*, and *PmWOXX* to the National Center for Biotechnology Information (NCBI), with the accession numbers PP033244, PP033245, and PP03324, respectively.

Afterward, SMART and Pfam analyses were conducted to validate the presence of conserved domains. Subsequently, we obtained a total of 11 *PmWOX* protein sequences, with their coding sequences provided in Appendix A. Among these 11 *WOX* proteins, the amino acid count ranged from 80 to 482, while the predicted molecular weight varied between 9.71 and 53.15 kDa. The pI values spanned from 5.45 to 10.26. Based on predictions by CELLO, Wolf PSORT, and Plant-mPLoc, all WOX genes were predicted to be located at the nucleus. Detailed information regarding the *PmWOX* genes can be found in Table 1.

### 2.2. Phylogenetic Analysis of PmWOX

The ML phylogenetic tree of 11 PmWOX protein sequences and other species’ WOX transcription factor families is shown in Figure 2. All 11 members in *P. massoniana* were divided into the WUS clade, the ancient clade and the intermediate clade. Results showed that only PmWOX6 was clustered in the intermediate clade. PmWOX1, PmWOX2, PmWOX4, and PmWOX8 were clustered in the ancient clade. PmWOX3, PmWOX5, PmWOX7, PmWOX9, PmWOXX, and PmWUS were clustered in the WUS clade. PmWOX1-4 and PmWUS are highly conserved in gymnosperm and angiosperm plants. However, PmWOX9 and PmWOXX are highly conserved in Pinaceae. All WOX members in each group were clearly divided into gymnosperm and angiosperm subgroups. It may indicate that WOX genes are structurally and functionally differentiated during plant evolution. We chose four conserved genes, PmWOX1-4, in plants for further study.

### 2.3. Protein Domain Analysis of PmWOX Proteins

According to the MEME program identification of the conserved motifs of the 11 PmWOX proteins (Figure 3), we found that all proteins had highly conserved motif 1. It indicates that motif 1 is the basic structure of WOX. It is essential for WOX to perform a function. The amino acid length of the 10 motifs ranged from 6 to 50 (Table 2). All members of the ancient clade contained motif 1, motif 2, and motif 5, except PmWOX8, which lacks motif 2. We found that motif 2, motif 4, and motif 5 disappeared during WOX evolution from the ancient clade to the modern clade. Motif 1 and motif 3 are distributed in all clades. Moreover, we found that all PmWOX genes contained a typical domain ‘helix-loop-helix-turn-helix’ (Figure 4) through multiple protein sequence alignment.

### 2.4. Analysis of the Transcriptional Profiles of PmWOX Genes under Drought Stress

The expression patterns under drought were validated by relative expression (qRT-PCR) and FPKM (RNA-seq) values. The heatmap of partial WOX genes was generated based on drought stress transcriptome data (Figure 5). The expression levels of some WOXs in the transcription group under drought stress were too low to be detected, so only six genes were given. The heatmap showed the expressions of partial genes under different field capacity levels. The expression of *PmWOX1* tended to decrease as the degree of drought deepened and increased in severe drought; all of them showed lower expression compared to CK. The expression of *PmWOX2* increased slightly in mild drought and decreased under medium and severe drought. The expression of *PmWOX3, PmWOX5*, and *PmWOX6* continued to increase as the drought deepened. On the contrary, the expression of *PmWOX4* was inhibited by drought. All of them showed lower expression compared to CK. Moreover, we selected four target genes for further verification by qRT-PCR. When *P. massoniana* was subjected to water loss conditions, the expression of *PmWOX2* decreased gradually and reached its lowest level at 7 days, followed by no significant change at 12 and 20 days. Conversely, the expression of *PmWOX1* and *PmWOX3* exhibited a significant increase at 12 days. Additionally, the expression of *PmWOX4* initially decreased within the first 12 days but peaked at 20 days. Due to the disparities in sample sources and sampling time between qRT-PCR and transcriptome sequencing, as well as the individual seedling used for qRT-PCR analysis compared to the pooled samples used in transcriptome sequencing, certain inconsistencies may arise. However, the expression trends of the four selected genes at T1, T2, and T3 are consistent with the trends at 3, 7, and 12 d.

### 2.5. Expression Patterns of PmWOX Genes in Different Tissues

The qRT-PCR results in Figure 6 showed the expression patterns of four *PmWOX* genes across eleven distinct tissues: shoot apices (T); young needle (YN); old needle (ON); young stem (YS); old stem (OS); xylem (X); phloem (P); root (R); callus (C); seed (S); young seedling (YSe). It is evident from the results that *PmWOX1* exhibits ubiquitous expression across all tissues, with particularly high levels observed in young needles and relatively low levels detected in roots, calluses, seeds, and young seedlings. Notably, *PmWOX2* displays significantly higher expression in callus compared to other tissues; moreover, its expression is almost undetectable in seeds and young seedlings. Among all tissues examined, *PmWOX3* demonstrates the highest expression levels in the callus followed by phloem and root; conversely, its expression is considerably lower in seeds and young seedlings. Similarly expressed across all tissues examined, *PmWOX4* exhibits notably high expression levels in needles, callus, and xylem, while displaying relatively low levels specifically within young seedlings; no significant differences were observed among other tissues. Overall, three genes exhibit elevated expressions within the callus except for *PmWOX1,* and all of them did not tend to be expressed in seeds and young seedlings.

### 2.6. Expression Patterns of PmWOX Genes under Different Treatments

The qRT-PCR results presented in Figure 7 demonstrate the expression patterns of four *PmWOX* genes under various treatments. Specifically, the expressions of *PmWOX1* and *PmWOX3* were induced significantly by ABA at 6 h; while *PmWOX2* and *PmWOX4* exhibited lower expression levels under ABA treatment. Furthermore, *PmWOX1*, *PmWOX2*, and *PmWOX4* can be induced at different time points by MeJA. However, MeJA inhibited the expression of *PmWOX3*. Notably, the expression of *PmWOX1* increased significantly at 6 h but decreased gradually thereafter under SA treatment. The expression of *PmWOX2* decreased significantly at 12 h, whereas that of *PmWOX3* peaked at 24 h. Under SA treatment, the expression level of only one gene (*PmWOX4*) continued to decrease over time. Additionally, ETH significantly inhibited both *PmWOX2* and *PmWOX3* expressions; however, other genes did not show sensitivity to ETH treatment. Moreover, H_2_O_2_ treatment led to a significant increase in the expression level of *PmWOX1* after 12 h, but caused a notable decrease in *PmWOX2* and *PmWOX3* expressions after this time point. Furthermore, the presence of PEG resulted in the inhibition of all four genes after 6 h. *PmWOX1*, *2*, and *4* were induced by NaCl, while *PmWOX3* did not show a significant change. In addition, *PmWOX1* and *PmWOX3* showed a continuous decrease in expression following injury to *P. massoniana*. The expression of *PmWOX2* and *PmWOX4* increased at 3 h but subsequently decreased.

### 2.7. Transcriptional Activation Activity

The yeast growth assay photographs (Figure 8) revealed that only yeast with pGBKT7-*PmWOX3* was able to grow on SD/-Trp/-His/-Ade medium and exhibited a blue color, while the other three failed to support yeast growth on SD/-Trp/-His/-Ade medium. These results unequivocally demonstrated that only *PmWOX3* exhibited transcriptional activation, whereas the remaining three genes did not exhibit any self-activation. The results suggested that *PmWOX3* can function as a transcriptional regulator in *P. massoniana*.

## 3. Discussion

The *WOX* family, known as plant-specific TFs, plays an important role in regulating plant growth, development, and abiotic stress. In recent years, the identification of *WOX* transcription factor families has been reported in various plants. However, most of the studies on the functions of the *WOX* genes have been focused on model plants like rice and *Arabidopsis* [3]. The identities and expression patterns of *WOX* genes involved in response to abiotic stresses in *P. massoniana* remain unknown. Therefore, we identified *WOX* genes in *P. massoniana,* and selected four genes for further functional analysis under different abiotic stresses.

In our study, we identified 11 *WOX* genes in *P. massoniana* and divided them into three clades, termed the WUS, intermediate, and ancient clade (Figure 2). Furthermore, the WUS clade had the largest number of members, similar to rice, *Arabidopsis* [3], poplar [11], and cotto [16]. The WUS clade exhibited the largest number of members in *P.massoniana* and other plant species, thereby confirming its high conservation as reported in previous studies. Moreover, we found that *PmWUS, PmWOX9*, and *PmWOXX* can be isolated from young seedlings rather than mature tissues. We hypothesized that, as the embryo matures into a seedling, their expression levels gradually decline and eventually become undetectable; they may perform specific functions in early development of *P. massoniana*. This is consistent with what has been reported in other plants. For example, *AtWOX5* is expressed in the quiescent center of the root tip, but shows low or undetectable expression levels in other regions [27]. During embryogenesis, *PaWOX2* exhibits the highest expression at the early development stages, but low levels were detected in seedling tissues [28]. *PpWOX3*, *PpWUS,* and *PpWOXX* were expressed in mature embryos. Specifically, *PpWUS* was expressed highly in 1–2 cm germinated embryos and lowly in 3 cm embryos. Compared to mature embryos, *PpWOX3* and *PpWOXX* show lower expression levels in germinated embryos [8]. The expression of *PpWUS*, *PpWOXX*, and *PpWOX3* is predominantly observed in the shoot apex of 3-week-old seedlings, while remaining undetectable in other tissues of the seedling.

The gene expression profile is correlated with their respective functions. We detected the expression levels of *PmWOX1-4* in different tissues. The results indicated their involvement in the growth and development processes at various stages in *P. massoniana*. The high expression observed in callus suggested that *PmWOX2* and *PmWOX3* may play an important role during the process of callus development, while also maintaining their involvement in the growth and development of *P. massoniana* throughout maturation. The transformation of gymnosperms poses significant challenges, primarily due to the difficulty in establishing a regeneration system. This is attributed to the challenging nature of callus induction, maintenance, and proliferation. The functions of *PmWOX2*, *PmWOX3*, *PmWOX9*, *PmWOXX*, and *PmWUS*, which were expressed in callus or at specific stages of embryo development, remain to be analyzed in the future to resolve this issue.

Proteins must be localized at the appropriate region to perform their function. Subcellular localization prediction results indicated that *WOX* TFs likely exert regulatory functions at the nucleus, which was consistent with *WOX* gene localization in *Arabidopsis* [29,30,31,32], rice [33,34], *Populus tomentosa* [35], and Rosaceae [36]. Regulatory responses to abiotic stress require several important metabolic proteins, such as osmoprotective proteins and regulatory proteins, as well as protein kinases and TFs involved in transduction in signal transduction pathways [37]. Some WOX genes can bind to stress-related genes to resist external stresses, such as *OsWOX13* [19] and *PagWOX11/12a* [20]. In addition, *PmWOX3* exhibited transcriptional activation and further exploration of its downstream regulated genes can provide insights into its molecular mechanisms involved in stress resistance and development of *P. massoniana*. For example, *OsWOX3* in rice can inhibit the expression of *YAB3* and regulate the development of rice leaves [38]. During early embryonic development, *AtWOX8*, together with its functional overlap gene *AtWOX9*, is activated by *WRKY2* transcription factor in the zygote, its basal daughter cell, and the hypophysis [39].

An increasing body of research demonstrates the involvement of WOX genes in the response to diverse abiotic stresses across various plant species [16,37]. WOX genes can show different responses when plants are under abiotic stresses. For, example, drought or salt stress can either up-regulate or down-regulate the transcriptional abundance of some WOX genes in soybean [37]. Similarly, we found that *PmWOX1* was significantly induced by ABA, MeJA, H_2_O_2_, and drought stress, indicating that *PmWOX1* may play an important role in responding to abiotic stresses. On the other hand, *PmWOX2* could respond to salt stress and MeJA treatment. *PmWOX3* showed significant induction during drought stress and did not exhibit sensitivity toward salt stress. Furthermore, while *PmWOX4* was non-sensitive to ETH, it could be induced by NaCl, injury, drought, and MeJA. *PmWOX1*, *PmWOX3*, and *PmWOX4* showed significant response to drought, as reported in rice [18], poplar [22], and cotton [26].

Numerous abiotic stresses lead to the generation of reactive oxygen species (ROS) in plants [40]. Excessive accumulation of ROS engenders oxidative stress, culminating in cell death and even plant death [41]. Consequently, plants undergo physiological and biochemical alterations that facilitate their manifestation of stress tolerance [42]. In summary, our findings suggest that WOX genes may play crucial roles in regulating *P. massoniana* under abiotic stress conditions. This study could provide valuable insights for subsequent functional gene studies. However, further genetic verification is warranted to elucidate their precise mechanisms.

## 4. Materials and Methods

### 4.1. Identification and Cloning of WOX Genes in P. massoniana

We acquired the Hidden Markov Model (HMM) profile of the WOX domain (PF00046) from the Pfam database (http://pfam.xfam.org/, accessed on 1 March 2023). The HMM profile was used to search the WOX proteins from four *P. massoniana* transcriptomes: CO2 stress transcriptome (PRJNA561037) [43], drought stress transcriptome (PRJNA595650) [44], young shoots transcriptome (PRJNA655997), and *P. massoniana* inoculated with the pine wood nematode transcriptome (SRA accession: PRJNA660087). A BLASTP search was performed against four transcriptomes using the Hidden Markov Model (HMM) profile. Transcription Factor Prediction (http://planttfdb.gao-lab.org/prediction.php, accessed on 5 March 2023) was used to predict WOX proteins. Then, we use Pfam and NCBI Conserved Domain Search (CD Search) (https://www.ncbi.nlm.nih.gov/Structure/cdd/wrpsb.cgi, accessed on 6 March 2023) to check the predicted WOX domain. Finally, certified sequences were selected after deleting sequences with more than 97% similarity. The primers of *PmWOX9*, *PmWUS*, and *PmWOXX* were designed by referring to *WOXX*, *WUS,* and *WOX3* sequences from *P. tabulaeformis*, *P. pinaster,* and *P. taeda* (Appendix A). The cDNA was obtained from 10-day-old *P. massoniana* seedlings. The PCR program for *PmWOXX* and *PmWOX9* is: (1) 94 °C for 30 s; (2) 98 °C for 10 s, 60 °C for 5 s, 72 °C 5 s, repeated 40 circles; (5) 72 °C for 5 min. The PCR program for *PmWUS* is: (1) 94 °C for 30 s; (2) 98 °C for 10 s, 62 °C for 5 s, 72 °C 5 s, repeated 45 circles; (5) 72 °C for 5 min. Each PCR mixture (20 µL) contained 1 µL of cDNA, 10 µL of 2× ApexHF FS PCR Master Mix (AG12202, Accurate Biology, Hunan, China), 1 µL of each primer, and 7 µL of ddH_2_O.

### 4.2. Bioinformatics and Phylogenetic Analysis of PmWOX Proteins

The basic information of PmWOXs was computed by the ProtParam tool (https://web.expasy.org/protparam/, accessed on 1 November 2023). The subcellular localization of PmWOX proteins was predicted through CELLO (http://cello.life.nctu.edu.tw/, accessed on 5 March 2023), WoLF PSORT (https://wolfpsort.hgc.jp/, accessed on 5 March 2023), and Plant-mPLoc (http://www.csbio.sjtu.edu.cn/bioinf/plant-multi/, accessed on 5 March 2023). We selected some WOX proteins from gymnosperms and angiosperms to construct a WOX phylogenetic tree. All protein sequences were downloaded from the NCBI database. Accession numbers (GenBank) of all members in the phylogenetic tree were listed in Appendix A. The number of each species are: 14 *P. pinaster* (Pp), 11 *P. taeda* (Pta), 12 *P. tabulaeformis* (Pt), 12 *Picea abies* (Pa), 7 *Ginkgo biloba* (Gb), 4 *P. sylvestris* (Ps); Angiosperm: 12 *Populus trichocarpa*, (Ptr), 21 *Zea mays* (Zm), 15 *Arabidopsis thaliana* (At), 13 *Oryza sativa* (Os). The phylogenetic tree of the amino acid sequences was constructed by the maximum likelihood (ML) method with MEGA-X software (v10.2.6) using 1000 bootstrap replicates. The phylogenetic tree was edited for visualization purposes with the online software EvolView (https://www.evolgenius.info/evolview/#login, accessed on 27 November 2023). The online program Multiple Expectation Maximization for Motif Elicitation (MEME) (http://meme-suite.org/tools/meme, accessed on 27 November 2023) was used to analyze the distribution of conserved motifs of PmWOX proteins and the number of motifs set to 10. The multiple sequences alignment of the conserved region was conducted by SnapGene (v6.0.2) using MUSCLE.

### 4.3. RNA-Seq Data Analysis under Drought Stress

One-year-old *P. massoniana* seedlings were obtained from the *P. massoniana* National Forest Seed Base in Duyun city, Guizhou Province, China. They were transplanted in a greenhouse for 3 months of adaptive growth in the Nanjing Forest University. A total of 120 healthy P. massoniana seedlings with the same height were selected in preparation for subsequent experiments. Soil water content was controlled by a weighing method. We set four gradients: normal water treatment (CK), mild (T1), medium (T2), and serious (T3). Each field capacity was set to: CK (80 ± 5)%, T1 (65 ± 5)%, T2 (50 ± 5)%, and T3 (35 ± 5)%. Each gradient had 10 samples repeated three times. The indoor temperature was controlled at 15–22 °C and the humidity was about 75%. All seedlings were grown under 16 h light culture (35,000 lx light intensity) and 8 h dark culture. Drought stress lasted for 60 days. Fragments per kilobase of the exon model per million reads mapped (FPKM) values were calculated to estimate the abundance of *PmWOX* gene transcripts. TBtools (Toolbox for Biologists) software (v2.034) was used to create heat maps of partial genes based on the values of log2 (FPKM + 1), and analyses were performed at the row scale.

### 4.4. Plant Materials and Abiotic Stress Treatments

Two-year-old *P. massoniana* seedlings were obtained from the State Key Laboratory of Tree Genetics and Breeding (Nanjing Forestry University). Callus was induced by a mature embryo of *P. massoniana*. Eight representative tissues (shoot apices; young needle; old needle; young stem; old stem; xylem; phloem and root) were collected from 3 two-year-old seedlings of the same height in the same pot. Whole young seedlings were collected 10 days after sowing. The two-year-old seedlings were treated with 9 abiotic stresses. The drought treatment was performed following the following method: collect samples five times in 20 days through natural evaporation after watering at 0 days, the field capacity being as follows: 0 d (67%); 3 d (63%); 7 d (58%); 12 d (46%); 20 d (34%). The mechanical damage treatment method was performed by cutting the upper half of the needles. The osmotic stress was induced by soaking the plants in the 15% polyethylene glycol (PEG6000) solution and 200 mM NaCl solution. For plant hormone treatment, the selected seedlings were sprayed independently with 100 µM ABA (abscisic acid); 1 mM SA (salicylic acid); 10 mM H_2_O_2_ (hydrogen peroxide); 10 mM MeJA (methyl jasmonate); 50 µM ETH (ethephon) solutions (50 mL) on the surface of needles. Afterward, needles were sampled at 0 h, 3 h, 6 h, 12 h, and 24 h after treatment, except in the case of drought stress. The needles under drought stress were sampled at 0 d, 3 d, 7 d, 12 d, and 20 d. All treatments were conducted on three biological replicates and three technological replicates.

### 4.5. RNA Extraction and qRT-PCR Analysis

Total RNA was extracted using the FastPure Plant Total RNA Isolation Kit (RC401, Vazyme Biotech, Nanjing, China). RNA concentration and purity were measured with a NanoDrop 2000 (Thermo Fisher Scientific, Waltham, MA, USA), and RNA integrity was estimated by 1% agarose gel electrophoresis. First-strand cDNA was synthesized using the One-step gDNA Removal and cDNA Synthesis Kit (AT311, TransGen Biotech, Beijing, China). Primers for quantitative real-time reverse transcription PCR (qRT-PCR) were designed using Primer 5.0 (Appendix A). SYBR Green reagents were used to detect the target sequence. Each PCR mixture (10 µL) contained 1 µL of diluted cDNA (20× dilution), 5 µL of SYBR Green Master Mix (11184ES03, Yeasen Biotech, Shanghai, China), 0.4 µL of each primer (10 µM), and 3.2 µL of ddH_2_O. The PCR program stages were: (1) 95 °C for 2 min (preincubation); (2) 95 °C for 10 s, (3) 60 °C for 30 s, repeated 40 times; the remaining steps used the instrument default settings. The PCR quality was estimated based on melting curves. *Alpha-tubulin* (*TUA*) gene was used as a reference gene [45]. Three independent biological replicates and three technical replicates for each biological replicate were examined. Quantification was achieved using comparative cycle threshold (Ct) values, and gene expression levels were calculated as 2 (^−∆∆Ct^) [∆CT = CT Target − CT *TUA*. ∆∆Ct = ∆Ct Target − ∆Ct CK]. Duncan’s test was used to examine the significance between different columns in IBM SPSS Statistics (Version 25). The results were marked with lowercase letters, starting with the largest average. The same lowercase letters between different columns indicate no significant difference. Completely different lowercase letters between different columns indicate a significant difference, *c* < 0.05. More than one lowercase letter in the same column indicate no significant difference between the column and other columns that contain one of the lowercase letters.

### 4.6. Transcriptional-Activation Activity Assay

We constructed recombinant vectors pGBKT7-*PmWOX* containing the complete open reading frame (ORF) sequences of *PmWOX1*, *PmWOX2*, *PmWOX3*, and *PmWOX4* to investigate their potential for transcriptional self-activation. The primer sequences used for vector construction are provided in Appendix A. The empty pGBKT7 vector was used as a negative control, and *PmC3H20* which was validated previously [46] was used as a positive control. Subsequently, these fusion vectors were transformed into yeast strain AH109 (YC1010, Weidi Biotechnology, Shanghai, China) according to the operational approach. Transformed yeast strains were screened on selective YPDA medium plates lacking tryptophan (SD/-Trp), and cultivated at 29 °C for 48 h. Single yeast colonies were collected in 10 µL ddH_2_O and confirmed by PCR analysis. Upon positive detection, the remaining positive bacterium liquid was diluted to a volume of 200 uL with ddH_2_O; subsequently, 5 μL of this dilution was plated onto different yeast media, lacking tryptophan/histidine (SD/-Trp/-His) and tryptophan/histidine/leucine (SD/-Trp/-His/-Ade), respectively. The later medium was supplemented with X-α-Gal on the surface to enable color development. Finally, photographic documentation was performed to record yeast growth.

## 5. Conclusions

In this study, we initially identified eight *PmWOX* genes in *P. massoniana,* and subsequently cloned three *WOX* genes based on sequences from other Pinaceae plants. Bioinformatics analysis was performed to further investigate their characteristics. Phylogenetic analysis revealed that all *WOX* members were categorized into three distinct clades, while the presence of a ‘helix-loop-helix-turn-helix’ structure was observed in all *WOX* genes. Notably, *PmWOX1*-*4* exhibit distinct responses to various stress conditions, whereas the expression of *PmWOX2*-*4* in callus suggested their potential roles in callus development. Furthermore, due to its transcriptional activation ability, *PmWOX3* could be considered a key gene for subsequent functional verification studies. Additionally, considering its significant expression in the callus, *PmWOX2* is an ideal candidate for investigating callus development specifically within *P. massoniana*. *PmWOX1* can be induced by ABA, ROS, and drought, suggesting its potential as a candidate gene for further investigation into its abiotic stress resistance. Importantly, this study represents the first report on *WOX* genes in *P.massoniana,* and provides valuable insights into the expression patterns of *PmWOX* genes. This study lays the foundation for future research on WOX genes, not only within *P. massoniana* but also across other related taxa.

## Figures and Tables

**Figure 1 ijms-25-01627-f001:**
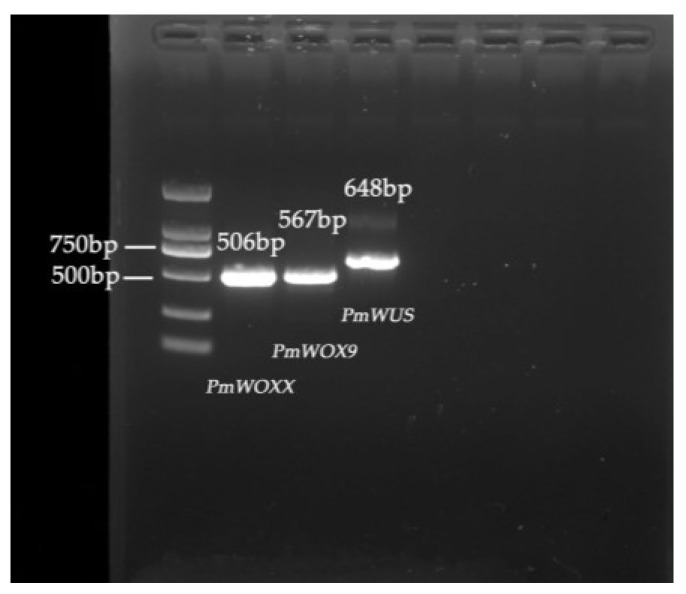
ORF amplification product of *PmWOX9*, *PmWOXX,* and *PmWUS*.

**Figure 2 ijms-25-01627-f002:**
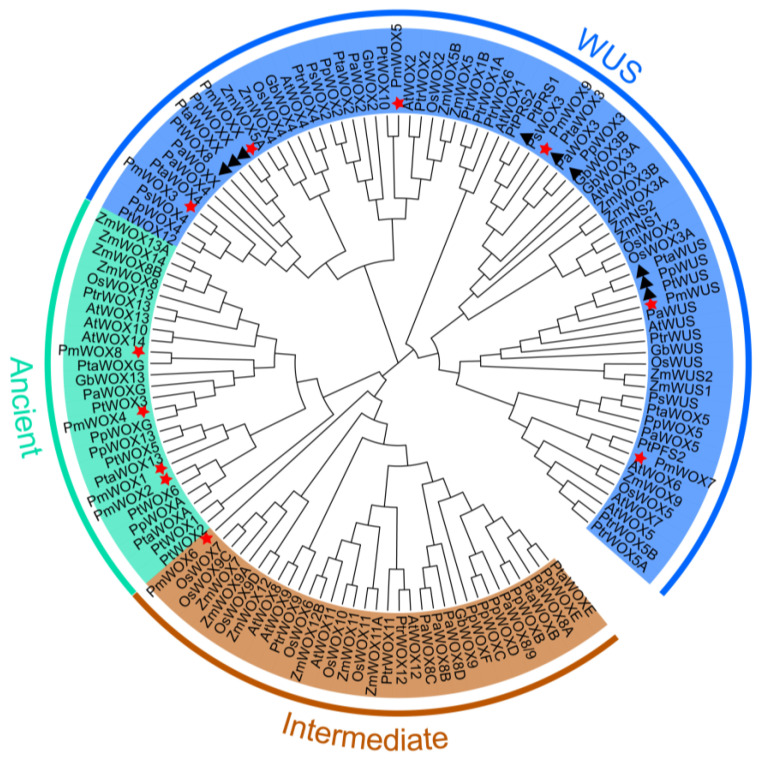
Phylogenetic tree of WOX gene family. Gymnosperm: *Pinus massoniana,* Pm; *Pinus pinaster*, Pp; *Pinus taeda*, Pta; *Pinus tabulaeformis*, Pt; *Picea abies*, Pa; *Ginkgo biloba*, Gb; *Pinus sylvestris*, Ps; Angiosperm: *Populus trichocarpa*, Ptr; *Zea mays*, Zm; *Arabidopsis thaliana*, At; *Oryza sativa*, Os. Red star represents WOX proteins in *P. massoniana*; Black triangle represents the homologous genes in *P. massoniana, P. taeda* and *P. tabulaeformis*.

**Figure 3 ijms-25-01627-f003:**
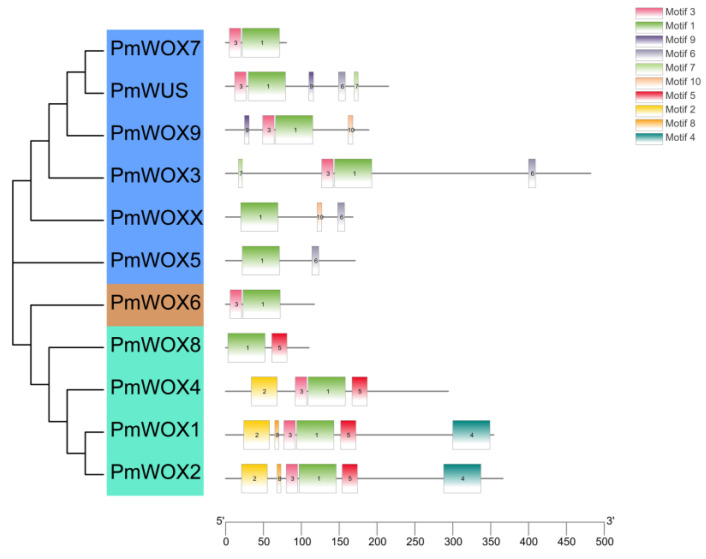
Prediction of conserved motifs of PmWOX proteins.

**Figure 4 ijms-25-01627-f004:**
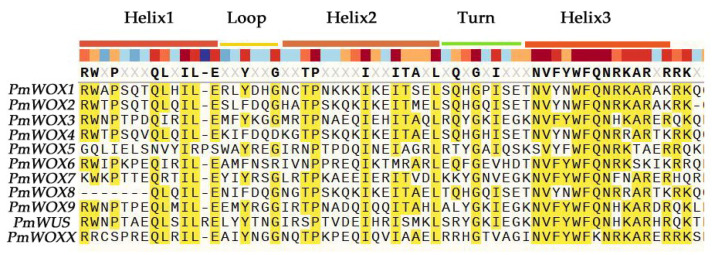
Alignment of the conserved region with PmWOX proteins.

**Figure 5 ijms-25-01627-f005:**
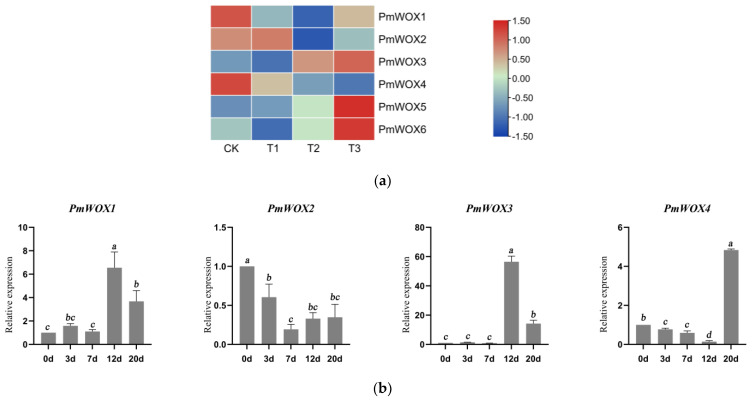
(**a**) Transcriptional profiles of WOX genes in *P. massoniana* under different drought levels: CK (80 ± 5)%, T1 (65 ± 5)%, T2 (50 ± 5)%, and T3 (35 ± 5)%. A heatmap was generated using log2 (FPKM + 1) values, then normalized by row scale. The color scale represents relative expression levels. (**b**) Relative expression of *PmWOX1*-*4* genes under drought stress. The same lowercase letters between different columns indicate no significant difference. The highest column is marked with ‘a’, then ‘b’ and so on. Completely different lowercase letters between different columns indicate a significant difference, *p* < 0.05. More than one lowercase letter in the same column indicates no significant difference between the column and other columns that contain one of the lowercase letters.” The relative expression in “0” was set as 1.

**Figure 6 ijms-25-01627-f006:**
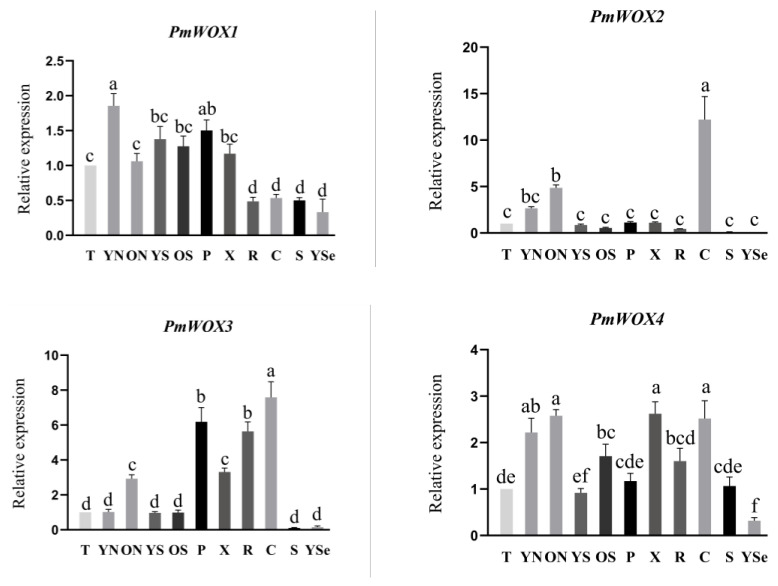
Relative expression of *PmWOX1*-*4* genes in different tissues. T: Shoot apices; YN: young needle; ON: old needle; YS: young stem; OS: old stem; X: xylem; P: phloem; R: root; C: callus; S: seed; YSe: young seedling. The same lowercase letters between different columns indicate no significant difference. The highest column is marked with ‘a’, then ‘b’ and so on. Completely different lowercase letters between different columns indicate a significant difference, *p* < 0.05. More than one lowercase letter in the same column indicate no significant difference between the column and other columns that contain one of the lowercase letters. The relative expression in “T”was set as 1.

**Figure 7 ijms-25-01627-f007:**
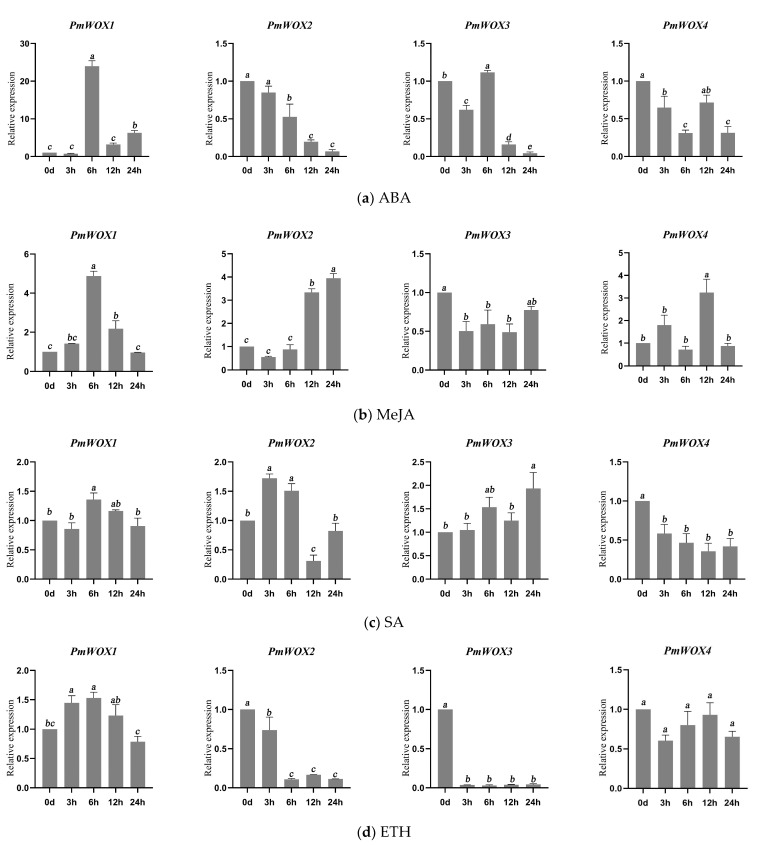
Relative expressions of *PmWOX* genes under different treatments. (**a**) ABA, (**b**) MeJA, (**c**) SA, (**d**) ETH, (**e**) H_2_O_2_, (**f**) PEG, (**g**) NaCl, (**h**) Injury. The same lowercase letters between different columns indicate no significant difference. The highest column is marked with ‘a’, then ‘b’ and so on. Completely different lowercase letters between different columns indicate a significant difference, *p* < 0.05. More than one lowercase letter in the same column indicate no significant difference between the column and other columns that contain one of the lowercase letters. The relative expression in “0” was set as 1.

**Figure 8 ijms-25-01627-f008:**
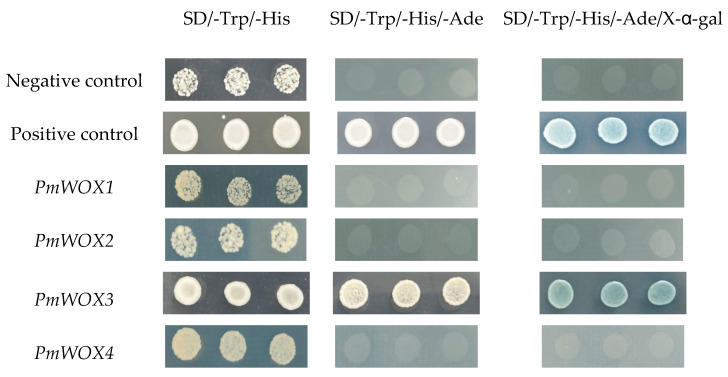
Transcriptional activation assay of four *PmWOX* genes. An empty pGBKT7 vector was used as a negative control. *PmC3H20* was used as a positive control.

**Table 1 ijms-25-01627-t001:** Detailed information and physicochemical properties of PmWOXs.

Protein ID	Prediction of Subcellular Localization	Amino Acid Number (aa)	Protein Molecular Weight (KD)	Isoelectric Point	Instability Index	Grand Average of Hydropathicity
PmWOX1	nucleus	354	40.03	6.21	47.92	−0.857
PmWOX2	nucleus	366	40.68	5.92	43.82	−0.783
PmWOX3	nucleus	482	53.15	7.11	57.57	−0.654
PmWOX4	nucleus	294	33.23	6.61	52.45	−0.657
PmWOX5	nucleus	171	19.64	8.80	38.32	−0.753
PmWOX6	nucleus	117	13.64	10.26	58.31	−0.997
PmWOX7	nucleus	80	9.71	9.35	52.38	−1.290
PmWOX8	nucleus	110	12.75	6.47	55.07	−1.286
PmWOX9	nucleus	189	21.95	8.19	63.33	−0.695
PmWUS	nucleus	215	24.88	8.05	63.64	−0.926
PmWOXX	nucleus	168	18.83	5.45	65.06	−0.656

**Table 2 ijms-25-01627-t002:** The sequences of conserved motifs.

Motif	Sequence	Number of Amino Acids in the Motif
1	ILEAJYDQGIRTPSKEQIKEITAELSQYGKIEGTNVFYWFQNRKARERRK	50
2	QVMTEEQLETLRRQICVYSTICSQLVEMHRAMSQQ	35
3	FRPSARTRWNPTPEQL	16
4	DGKQWZVPVGVVDVRRMFGENAVLLDSRGHMVPTNDMGMSFHPLQGSEGY	50
5	GESEVDTDLESPKEKKVKMDH	21
6	TLZLFPVHPE	10
7	CCWGGC	10
8	RMLYDL	6
9	CEEAFCC	7
10	CVEEYDC	7

## Data Availability

The data presented in this study are available in Appendix A.

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
