# Peer review of "Identification and Expression Patterns of WOX Transcription Factors under Abiotic Stresses in Pinus massoniana"

_ijms, 2024, doi:10.3390/ijms25031627_

Round 1

Reviewer 1 Report

Comments and Suggestions for Authors

Based on transcriptome data, the authors attempted to identify the WOX transcription factors under abiotic stresses in Pinus massoniana. Their expression pattern also were performed under different abiotic stresses. The authors have done a lot of works, but the research still remain some problems in experimental design, data analysis and writing, as follows:

1.     Abstract.  Sum up your results and tell us the conclusion. A clear and concise abstract is better.

2.     The main purpose of the article is to identify WOX transcription factors under abiotic stress, and the introduction should focus on introducing the biological functions of WOX transcription factors under different abiotic stresses. Some description logic in Introduction is confusing.

3.     In this manuscript, only four transcriptome data were used to identify the WOX transcription factors under abiotic stress. However, only two transcriptome data were under abiotic stress (CO2 and drought), which is less to represent the abiotic stress. Additionally, we know that transcriptome data is gene expression data in a specific time and space. Can all WOX genes be identified only using transcriptome data? WOX transcription factors are involved in many biological processes, and only 2 stresses can identify all WOX genes?

4.     The author amplified three additional WOX genes from seedlings using PCR cloning, but the template used was the cDNA reversed from RNA. Similarly, will partial expression of gene RNAs based on specific time and space lead to the omission of WOX genes?

5.     The assembly of transcriptome data without reference genome can easily produce a large number of duplicated or incomplete transcripts. Will such duplicated or incomplete transcripts lead to the identified WOX gene being duplicated or incomplete?

6.     According to Fig2, PmWOX5 is mainly clustered with other species WOX2, while PmWOX7 is clustered with other species WOX5. Why?

7.     The visualization of the data or figures in the article needs to be improved, and the color scheme of the images should be as uniform as possible;

8. There are many grammatical errors in the manuscript, the logics of some sentences are confusing and does not meet the writing standards. English may need to be checked throughout the text by a native English speaker.

Comments on the Quality of English Language

There are many grammatical errors in the manuscript, the logics of some sentences are confusing and does not meet the writing standards. English may need to be checked throughout the text by a native English speaker.

Reviewer 2 Report

Comments and Suggestions for Authors

1.       Line 52, “contains” should be not italic.

2.       The third and fourth paragraphs in the introduction part should be rewritten.

3.       Line 90, “designated them as PmWOX1-8”, what’s the naming rules, the authors should explain.

4.       Line 99, one blank should be inserted between numbers and ‘bp’.

5.       The coding sequences of PmWOX9, PmWUS and PmWOXX should be uploaded to public database and the deposit ID should be provided.

6.       For the “Phylogenetic analysis of PmWOX” part, all the names of WOX are protein, and italic should be change to regular type.

7.       For table 2, the annotations for each motif should be provided.

8.       Line 336 to 340, the authors should describe how did they obtained these protein sequences.

9.       The conclusion part should be also rewritten. Current version just describes their results again.

Reviewer 3 Report

Comments and Suggestions for Authors

In the submitted manuscript by Kongshu Ji and colleagues entitled “Identification and expression patterns of WOX transcription factors under abiotic stresses in Pinus massoniana”, the authors reveal the role of WUSCHEL-related homeobox (WOX) transcription factors in plant development regulation and under abiotic stress. They identified 11 WOX, employed phylogenetic tree analysis, and tested their expression. Overall, the manuscript is well-written, and the figures have a good presentation. However, there is issues that should be carefully addressed.

In Results section

Subsection 2.4. You mention that 6 genes (WOX1-6), were able to be detected for their expression under abiotic stresses, which is understandable. Then in the rest of subsections 2.5., 2.6. and 2.7. You had analyzed only 4 genes. Please provide an explanation why you exclude WOX5 and WOX6 from the rest of the analysis.

Round 2

Reviewer 2 Report

Comments and Suggestions for Authors

1.       For ‘2.4. Analysis of the transcriptional profiles of PmWOX genes’, the authors only present the drought responsive genes in the Figure 5. In this case, I think the title of this part should be changed.

2.       For Figure 5 and (i) of Figure 7, both results were presented the drought responsive PmWOX genes. I strongly suggest the authors to combine two results.

3.       Line 332 to 334, for the color of the heat map, which one is right, based on the ‘log2(FPKM+0.01) values’ or ‘the log2 fold change’? If the fold change were used, all the color for the CK should be same. Further, how did the author get the FPKM value from the original reads, the authors also should describe the methods in the part of  ‘Materials and Methods’.

4.       For the expression patterns of PmWOX genes in different tissues and under different treatments, why did the authors only present the expression patterns of PmWOX1-4, how about other genes, no expression?

5.       I'm just curious. For the sampling of different treatments, why did the authors choose the same sample time point?
